# Life Table Parameters of *Tetranychus merganser* Boudreaux (Acari: Tetranychidae) on Five Host Plants

**DOI:** 10.3390/insects14050473

**Published:** 2023-05-17

**Authors:** Ma. Teresa de Jesús Segura-Martínez, Salvador Ordaz-Silva, Agustín Hernández-Juárez, Rapucel Tonantzin Quetzalli Heinz-Castro, Sandra Grisell Mora-Ravelo, Julio César Chacón-Hernández

**Affiliations:** 1Faculty of Engineering and Sciences, Universidad Autónoma de Tamaulipas, Ciudad Victoria 87120, Tamaulipas, Mexico; tsegura@docentes.uat.edu.mx (M.T.d.J.S.-M.); sgmora@docentes.uat.edu.mx (S.G.M.-R.); 2Faculty of Business and Engineering San Quintín, Universidad Autónoma de Baja California, San Quintín 22930, Baja California, Mexico; salvador.ordaz.silva@uabc.edu.mx; 3Parasitology Department, Universidad Autónoma Agraria Antonio Narro, Saltillo 25315, Coahuila, Mexico; chinoahj14@hotmail.com; 4Faculty of Agronomy and Veterinary, Universidad Autónoma de San Luis Potosí, Soledad de Graciano Sánchez 78321, San Luis Potosí, Mexico; rapucel.heinz@uaslp.mx

**Keywords:** demographic parameters, development, fecundity, red spider mite, survival

## Abstract

**Simple Summary:**

In Mexico, the red spider mite, *Tetranychus merganser*, is one of the most economically important pest mites in papaya crops, causing damage to the fruits and leaves. This research aimed to assess the effect of five host plants (*Carica papaya* L. (papaya), *Capsicum annuum* var. *glabriusculum* (Dunal) Heiser and Pickersgill (chili piquin), *Helietta parvifolia* (Gray) Benth. (barreta), *Phaseolus vulgaris* L. (bean), and *Rosa hybrida* L. (rosebush)) on the biology and life table parameters of *T. merganser*. We hypothesized that *T. merganser* has a better biological performance on papaya than on other host plants. The results of this research can be used to develop management and control strategies for *T. merganser*. The mean immature period of red spider mite females was longer on barreta leaf disks than on chili piquin, rosebush, papaya, and bean. The oviposition of *T. merganser* was greater on bean disks than on the other host plants. Host plants affect the number of eggs laid by the red spider mites, which reduce their intrinsic rate of natural increase. The demographic parameters suggest the unsuitability of barreta as the host for the development of red spider mites, and the best performance of *T. merganser* was on *P. vulgaris*.

**Abstract:**

The quality of the host plant affects the life history parameters of tetranychid mites. The biology and fertility life tables of *Tetranychus merganser* on five host plants (*Carica papaya*, *Phaseolus vulgaris*, *Capsicum annuum* var. *glabriusculum*, *Helietta parvifolia*, and *Rosa hybrida*) were assessed under laboratory conditions at 28 ± 1 °C and 70–80% relative humidity (RH) with a photoperiod of 12:12 h (L:D). The development period of immature females differed among the tested host plants and ranged from 9.32 days on *P. vulgaris* to 11.34 days on *H. parvifolia*. For immature males, it ranged from 9.25 days on *P. vulgaris* to 11.50 days on *H. parvifolia*. The female survival rate varied from 53.97% on *H. parvifolia* to 94.74% on *P. vulgaris*. The highest total fecundity rate was recorded on *P. vulgaris* (125.40 eggs/female) and the lowest on *H. parvifolia* (43.92 eggs/female). The intrinsic rate of increase (rm) varied from 0.271 (*H. parvifolia*) to 0.391 (*P. vulgaris*). The net reproductive rate (R_O_) was higher on *P. vulgaris* than on the other host plants. The longest mean generation time (GT) was calculated on *C. annuum* var. *glabriusculum* and the shortest on *Rosa hybrida*. The demographic parameters suggest the unsuitability of *H. parvifolia* as the host for the development of red spider mites, and the best performance of *T. merganser* was on *P. vulgaris*.

## 1. Introduction

The red spider mite, *Tetranychus merganser* Boudreaux (Acari: Tetranychidae), feeds on twenty-three host plants [1,2,3,4,5], including wild chili pepper [*Capsicum annuum* L. var. *glabriusculum* (Dunal) Heiser y Pickersgill (Solanaceae)], papaya [*Carica papaya* L. (Caricaceae)], rose bushes [(*Rosa* sp. L. (Rosaceae)], and prickly pear cactus [(*Opuntia ficus*-*indica* L.) Miller (Cactaceae)]. It is found in China, Mexico, the United States, and Thailand [5]. The red spider mite is considered an emerging pest in Mexican agriculture due to the recent increase in the range of its host plants and geographic expansion [6]. The feeding habits of *T. merganser* destroy the epidermal tissue, the parenchyma, and the chloroplasts of the leaves, causing the host plant to fail to grow, develop, and reproduce [7,8]. Control of *T. merganser* is through chemical insecticides. However, the red spider has a short life cycle and high reproductive potential, and the excessive use of acaricidal applications can cause them to develop resistance to these compounds [9,10]. Other preventive control methods for *T. merganser* have been reported. They cause a minimal environmental impact, including predator mites (*Neoseiulius californicus* McGregor and *Amblyseius swirskii* Athias-Henriot (Gamasida: Phytoseiidae)) [8], botanical extract (*Moringa oleifera* ethanol extract) [11], and entomopathogenic fungi (*Metarhizium anisopliae* s.l. and *Beauveria bassiana*) [6].

*T. merganser* biology needs to be better understood since it has only been studied on *Phaseolus vulgaris* L. (Fabaceae) [9] and *C. papaya* [12], at different temperatures. These authors found that the biological and demographic parameters of red spider mites, such as survival, developmental period, total fecundity, and intrinsic rate of natural increase, differ in response to different temperatures. Ullah et al. [9] reported that the optimal development for red spider mites was at 30 °C on *P. vulgaris* at different tested temperatures (15 to 37.5 °C and 60–70% relative humidity). Furthermore, Reyes-Pérez et al. [12] found that *T. merganser* had better performance at 27 °C on *C. papaya*, when evaluated in a range of 19 to 35 °C and 60 ± 2% relative humidity. Treviño-Barbosa et al. [13] registered the number of eggs, survival, and percentage of feeding damage daily for four days, and on the fourth day they calculated the growth rate of *T. merganser* on *C*. *papaya*, *P. vulgaris*, *C. annuum* var. *glabriusculum*, *Moringa oleifera* Lam. (Moringaceae), *Helietta parvifolia* (Gray) Benth. (Rutaceae), *Pittosporum tobira* (Thunb.) W.T. Aiton (Pittosporaceae), and *Thevetia ahouai* L. A. DC. (Apocynaceae). They found that the red spider mite has better performance on *C. papaya* than on other plant species. According to Treviño-Barbosa et al. [13], the population growth parameters of *T. merganser*, such as fecundity, survival, and food intake, may vary in response to the changes in temperature, host plant species, and nutrition quality of plants [9,12,13]. However, Treviño-Barbosa et al. [13] have already reported some demographic and biological parameters of *T. merganser*, such as the intrinsic (infinitesimal) rate of increase (r), the finite growth rate (λ), the doubling time (DT), and the oviposition on *C. papaya*, *P. vulgaris*, *C. annuum* var. *glabriusculum*, and *H. parvifolia*. These results approximate the population growth of the red spider mite when it feeds on a host plant since r is a crude rate, and they establish a foundation for examining population growth patterns over short periods [14,15], such as in the study by Treviño-Barbosa et al. [13]. They calculated r on the fourth day. To have a clear and systematic picture of the specific age at birth (mx) and survival (lx) of the red spider mite population [15,16], we analyzed in more detail the developmental time from egg to adult for both females and males; the pre-oviposition, oviposition, and post-oviposition periods; the fecundity rate; survival rates of all immature stages and differences between male and female; and several life table parameters of *T. merganser*.

Life table parameters have been used as an indicator to assess population growth under a given set of conditions, thus showing the effect of different host plants or varieties on the mite [17]. Furthermore, the mite fertility life table parameters are used as an endpoint to reveal the susceptibility or resistance of host plants to the mite, since host plants that support low mite population growth are important for integrated pest management [17]. The aim of this research was to assess the effect of five host plants (*C*. *papaya*, *C. annuum* var. *glabriusculum*, *H. parvifolia*, *P. vulgaris*, and *Rosa hybrida*) on the biology and life table parameters of *T. merganser*. Our hypothesis was that *T. merganser* has a better biological performance on *C. papaya* than on other host plants. The results of this research can be used to develop management and control strategies for *T. merganser*.

## 2. Materials and Methods

### 2.1. Red Spider Mite Rearing

Individuals of different stages of development (larvae, nymphs, and adults) and the sex of *T. merganser* were collected from piquin pepper (*C. annuum* var. *glabriusculum*) in “Cañón de la Peregrina” (23°46′41″ N, 99°12′12″ W, 411 m above sea level), located in the protected natural area “Altas Cumbres” in the Victoria municipality, Tamaulipas, Mexico. The mites were reared on bean plants (*P. vulgaris*) grown in plastic pots (15 cm diameter × 10 cm height) under greenhouse conditions (30 ± 2 °C and 70 ± 10% relative humidity (HR)) for three months (several generations) before carrying out the experiments.

### 2.2. Host Plants

For this study, five host plants of *T. merganser* were selected, i.e., *C. papaya*, *P. vulgaris*, *H. parvifolia*, *C. annuum* var. *glabriusculum*, and *Rosa* sp. (Table 1).

We collected mature leaves from each species of host plants under field conditions (Table 1). These leaves did not show any damage or any symptom of the presence of fungi or bacteria. The leaves of each host plant were transported in resealable plastic bags inside a cooler with a frozen gel pack at a temperature of 5 ± 2 °C to the Population Ecology Laboratory of the Institute of Ecology of the Autonomous University of Tamaulipas. The transfer time of the leaves to the laboratory depended on the location of the host plants, but ranged from 5 to 30 min. In the laboratory, the leaves were treated with a 2 min wash with 1.5% sodium hypochlorite solution and cut into 4 cm^2^ squares [13].

### 2.3. Immature Development and Performance of Adults

All the experiments were carried out under laboratory conditions at 28 ± 1 °C and 70–80% relative humidity (RH) with a photoperiod of 12:12 h (light: dark). This study was conducted in the Population Ecology Laboratory of the Institute of Applied Ecology, Autonomous University of Tamaulipas, during 2020–2022.

To determine the development times, survival rate, longevity, and fecundity of *T. merganser*, we used the methodologies of Uddin et al. [17] and Gotoh and Gomi [18]. A pair of adult mites (one newly emerged female and one male) were randomly selected from the stock rearing of *T. merganser* and placed on the leaf squares of each host plant with a fine camel hairbrush. With the help of a sterile scalpel, we cut the leaf squares of each host plant to 4 cm^2^. Each leaf square was placed on water-saturated cotton in a 5 cm diameter Petri dish, with the underside of the leaf facing up. We let the female and male mate for a period of 6 h. During this period, we monitored the oviposition, and immediately a single egg was left on each leaf square (females, males, and additional eggs were removed). During all experiments, leaf squares were changed after 3–4 days to ensure freshness, and individuals were transferred to new squares. Observations were made twice daily with the help of a stereoscopic microscope (UNICO Stereo & Zoom Microscopes ZM180, Dayton, NJ, USA). The duration of the development time from egg to adult, the survival, and the sex ratio (% females) of the emerging mites were recorded for each host plant and determined after reaching adulthood.

### 2.4. Oviposition and Life Table Parameters

When the red mite female reached the teleiochrysalis stage, one female and one male were placed on the same leaf square for mating. We kept the male on the leaf square for as long as the female was alive. If a male died before the female or got caught in the cotton threads, he was replaced by a young one. Females that became entangled in the cotton thread or killed due to improper handling or drowned were excluded from the data analysis. The eggs laid by a female were recorded daily until her death. As the leaf square aged, the mites transferred to new leaf squares. In this way, the oviposition period, the total number of eggs laid per female, the eggs laid per female per day, the post-oviposition period, and the longevity of the female of 25 *T. merganser* females per host plant were determined [16,17].

We used the daily age-specific survival rate (l_x_) and age-specific fecundity (m_x_) to generate the life tables of *T. merganser* on each host plant. The intrinsic rate of natural increase ($r_{m}$) was estimated from the fertility table according to the equation given by Carey [14] and Birch [19]: ∑e^−rx^ l_x_ m_x_ = 1. We calculated the gross reproductive rate (GRR = ∑m_x_), net reproductive rate (R_O_ = ∑l_x_m_x_, the mean generation time (T = lnR_O_/rm), growth capacity rate (rc = log_e_R_O_/T), the finite rate of increase (λ = e^rm^), and the doubling time (DT = ln2/rm) based on Birch [19].

### 2.5. Statistical Analyses

Significant differences in life table parameters (GRR, rm, rc, R_O_, GT, λ y DT) were tested using the Jackknife procedure to estimate the standard error of demographic parameters [20,21]. The effects of host plants on the development, oviposition period, longevity, and fecundity of *T. merganser* were separately analyzed by a one-way ANOVA, followed by the Tukey HSD test (*p* < 0.05) using R software version 4.2.1. [22].

## 3. Results

### 3.1. Development and Survival of Immature Stages

Both *T. merganser* females and males successfully completed their development on the five tested host plants. The developmental time of the egg, larva, protochrysalis, protonymph, deutochrysalis, deutonymph, and theliochrysalis of the females differed significantly between the tested host plants (*p* < 0.05; Table 2). The total developmental time (from egg to adult) for females differed significantly between host plants (*p* < 0.0001). *T. merganser* females developed faster on *P. vulgaris* (9.32 days) and *C. papaya* (9.58 days) than on *R. hybrida* (10.30 days), *C. annuum* var. *glabriusculum* (11.30 days), and *H. parvifolia* (11.34 days) (Table 2).

Regarding the *T. merganser* male, the mean developmental time of the egg, larva, protonymph, deutochrysalis, deutonymph, and theliochrysalis differed significantly between the tested host plants (*p* < 0.05; Table 3). However, the protochrysalis period was not significant (*p* > 0.05). The total developmental time (from egg to adult) of the males differed significantly between host plants (*p* < 0.0001). Red spider mite males developed faster on *P. vulgaris* (9.25 days) and *C. papaya* (9.71 days) than on *Rosa* sp. (10.22 days), *C. annuum* var. *glabriusculum* (11.21 days), and *H. parvifolia* (11.50 days) (Table 3).

The survival of *T. merganser* females and males showed that the mites successfully developed on the five host plants (Table 4 and Table 5). The survival rate of the immature stages (from egg to adult) of *T. merganser* females varied significantly from 53.97% on *H. parvifolia* to 94.74% on *P. vulgaris* (Table 4). Nevertheless, the survival rate of males did not differ significantly between the host plants (Table 5). Although the survival rates of females and males differed between host plants (Table 4 and Table 5), Pearson’s association test did not show a significant relationship between sex and host plants (χ^2^ = 1.410; df = 4; *p* = 0.8424). Sex ratios (%female) ranged from 77.27 on *H. parvifolia* to 79.07 on *P. vulgaris*. However, this was statistically similar between the five host plants (χ^2^ = 0.0769, df = 4, *p* = 0.9993).

### 3.2. Adult Longevity and Oviposition

Host plants had a significant effect on the period of pre-oviposition, oviposition, and post-oviposition, as well as the longevity and fecundity of *T. merganser* adult females (*p* < 0.0001; Table 6). The pre-oviposition period of *T. merganser* was longer on *C. papaya*. The highest value of the oviposition period was found on *C. annuum* var. *glabriusculum* (18.24 days) and *R. hybryda* (17.44 days), and the shorter value on *H. parvifolia* (13.12 days). Nevertheless, the female longevity ranged between 15.04 (*H. parvifolia*) and 20.84 (*C. annuum* var. *glabriusculum*) days and was significantly influenced by the host plant (*p* < 0.0001). Both the total fecundity and number of eggs laid by females were significantly higher on *P. vulgaris* (125.40 and 7.81, respectively) (Tukey’s tests, *p* < 0.05; Table 6).

### 3.3. Life Table Parameters

The gross reproduction rate (GRR), net reproductive rate (R_O_), growth capacity rate (rc, day^−1^), intrinsic rate of natural increase (rm, day^−1^), mean generation time (GT, in days), finite rate of increase (λ), and doubling time (DT) of *T. merganser* on the five host plants are shown in Table 7. The life table parameters GRR, R_O_, rm, GT, λ, and DT values were significantly different between the five host plants (*p* < 0.0001). The GRR of *T. merganser* differed significantly between host plants, and it was highest on *P. vulgaris* (109.364) and lowest on *H. parvifolia* (48.039). The rm value of *T. merganser* was significantly higher and lower than on *P. vulgaris* (0.391) and *H. parvifolia* (0.271), respectively. Similarly, R_O_, rc, and λ were the highest when *T. merganser* fed on *P. vulgaris* (75.240, 0.642, and 1.478, respectively) and the lowest when it fed on *H. parvifolia* (22.840, 0.439, and 1.311, respectively). The mean generation time (GT) differed among the host plants and was longest on *C. annuum* var. *glabriusculum* (12.037) and shortest on *R. hybrida* (10.817). The mean doubling time was also found to be significantly different on the host plants (*p* < 0.0001). The highest and lowest DT values were observed on *H. parvifolia* (2.561) and *P. vulgaris* (1.774), respectively (Table 7).

## 4. Discussion

This study showed that *T. merganser* can survive successfully and complete its development on *P. vulgaris*, *C. papaya*, *R. hybrida*, *C. annuum* var. *glabriusculum*, and *H. parvifolia*, and these plants significantly affected its life history parameters. The life table parameters are reliable tools to assess the quality of host plant effects on the biology and fertility of phytophagous arthropods, since they indicate their population growth rates in the current and next generations. Therefore, understanding them is essential to develop an integrated pest management strategy [23,24].

In the literature, there are few references to the developmental time (from egg to adult) of *T. merganser* on different host plants [9,12]. However, other studies have researched the developmental time of *Tetranychus* spp., and their findings are very similar to ours. This research showed that the duration of development from egg to adult for females and males ranged from 9.32 to 11.34 and 9.25 to 11.50 days on *P. vulgaris* and *H. parvifolia* at 28 °C, respectively. The developmental times of female and male red spider mites were reported to be 8.80 days and 8.3 days on *P. vulgaris* at 27.5 °C, respectively [8]. Islam et al. [25] reported that the mean immature period of the *T. truncatus* Ehara (Acari: Tetranychidae) female was longer on *Corchorus capsularis* L. (Malvacea) (7.40 ± 0.07) than on *Lablab purpureus* (L.) Sweet (Fabaceae) (7.00 ± 0.04 days) and *C. papaya* (6.90 ± 0.04 days). Puspitarini et al. [26] reported that *T. urticae* Koch (Acari: Tetranychidae) developed faster when fed *Fragaria x ananassa* (Duchesne ex Weston) Duchesne ex Rozier (9.03 ± 0.40 days) than *Chrysanthemum indicum* L. (Asteraceae) (10.05 ± 0.90 days) and *C. papaya* (12.37 ± 2.40 days). Furthermore, Draz et al. [27] documented that the development times of the immature stages of *T. urticae* were longer on *C. annuum* (11.90 ± 0.41 days) and *Solanum melongena* L. (Solanaceae) (11.30 ± 0.40 days) and shorter on *Cucumis sativus* L. (Cucurbitaceae) (10.50 ± 0.27 days) and *Citrullus lanatus* (Thunb.) Matsum. and Nakai (Cucurbitaceae) (9.50 ± 0.22 days). De Lima et al. [28] found that development from the egg to adult female of *Tetranychus bastosi* Tuttle, Baker and Sales (Acari: Tetranychidae) varied significantly by the host plant (10.5 ± 0.29 days on *P. vulgaris*, 11.2 ± 0.18 on *C. papaya*, and 12.3 ± 0.24 days on *Manihot esculenta* Crantz (Euphorbiaceae)). Adango et al. [29] documented that the period of development from egg to adult of the *T. ludeni* Zacher (Acari: Tetranychidae) female at 27 °C was shorter on *Amaranthus cruentus* L. (Amaranthaceae) (9.6 days) than on *Solanum macrocarpon* L. (Solanaceae) (10.1 days). Barroncas et al. [30] found that *T. mexicanus* McGregor (Acari: Tetranychidae) developed faster on *C. papaya* (11.2 ± 0.07 days) than on *Passiflora edulis* Sims (Passifloraceae) (11.9 ± 0.13 days). Islam et al. [25] and Puspitarini et al. [26] mentioned that the development time from egg to adult of *Tetranychus* spp. was dependent on host plants. In addition, the genetic background of the tested mite population, quality of host plants used for feeding the mite, and laboratory conditions affect the development period of the mite [26].

In this study, the survival rate trend of the *T. merganser* female in relation to the host plants was as follows: *P. vulgaris* > *C. papaya* > *R. hybrida* > *C. annuum* var. *glabriusculum* > *H. parvifolia*. Although the survival rates of *T. merganser* males were higher on *P. vulgaris* (81.82%) than on *C. papaya* (70.00%), *R. hybrida* (63.34), *C. annuum* var. *glabriusculum* (60.87%), and *H. parvifolia* (50.00%), they did not differ significantly. Ullah et al. [8] reported an 87.5% survival rate for *T. merganser* (female + male) when the mites fed on *P. vulgaris*. Islam et al. [24] reported that the survival from egg to adult of *T. truncatus* was greater on *L. purpureus* (95.9%), *C. capsularis* (93.4%), and *C. papaya* (91.8%). Draz et al. [27] documented 100% *T. urticae* survival when the mites fed on *C. lanatus*, and only 53.33%, 43.75%, and 42.86% survived when they fed on *S. melongena*, *C. sativus*, and *C. annuum*, respectively. Barroncas et al. [30] found that the survival rate of *T. mexicanus* differed significantly between *C. papaya* (92.0 ± 0.04%) and *P. edulis* (79.0 ± 0.06%).

Different host plants significantly affected the total fecundity and daily oviposition of *T. merganser* females. The fecundity of *T. merganser* was significantly higher on *P. vulgaris* (125.40 ± 1.67) than on *C. papaya* (99.60 ± 1.24), *C. annuum* var. *glabriusculum* (64.65 ± 0.87), *R. hybrida* (62.10 ± 1.21), and *H. parvifolia* (43.92 ± 0.07). Ullah et al. [9] reported that the fecundity of *T. merganser* was 143.9 (± 10.49) and 146.2 (± 6.35) eggs on *P. vulgaris* at 28 and 30 °C, respectively, which are greater than that reported in this study. Reyes-Pérez et al. [12] documented that the total fecundity of *T. merganser* was 70.73 and 52.45 eggs on *C. papaya* at 23 and 27 °C, respectively. Treviño-Barbosa et al. [13] documented that the mean number of *T. merganser* eggs laid per female on *C. papaya* (9.39 ± 0.41) was higher than on *P. vulgaris* (5.54 ± 0.37), *M. oleifera* (3.97 ± 0.16), *C. annuum* var. *glabriusculum* (2.58 ± 0.14), *H. parvifolia* (1.57 ± 0.05), *P. tobira* (1.00 ± 0.04), and *T. ahouai* (1.06 ± 0.05). *Tetranychus* spp. have shown variations in fecundity when reared on different host plants. Islam et al. [25] reported that the fecundity of *T. truncatus* was higher on *C. capsularis* (126.9 eggs) than on *L. purpureus* (86.5 eggs) and *C. papaya* (84.2 eggs). *Tetranychus mexicanus* oviposited 106.0 (±8.96) and 81.7 (±7.21) eggs when fed on *C. papaya* and *P. edulis*, respectively [30]. The fecundity of *T. turkestani* was 65.13 (±3.68), 44.03 (±3.81), and 32.69 (±2.82) eggs on *Vigna unguiculata* (L.) Walp. (Fabaceae), *Phaseolus lunatus* L. (Fabaceae), and *Phaseolus calcaratus* Roxb. (Fabaceae) [31]. Although we did not examine the nutritional quality, secondary metabolites, and morphological characteristics of the tested host plant leaves, reports suggested that these factors affect the development and reproduction for several tetranychid species [25,26,32,33,34], as occurred in this study, where the fecundity of *T. merganser* females was higher on *P. vulgaris* than on other plant species. Papp et al. [32] reported that high nitrogen concentrations stimulated the egg production of the European red mite, *Panonychus ulmi* Koch (Acari: Tetranychidae). Karley et al. [33] found a negative relationship between the number of eggs laid by *T. urticae* and the number of trichomes on the leaves of different varieties and accessions of red raspberry, *Rubus idaeus* L. (Rosaceae). Furthermore, Vásquez et al. [34] documented a negative relationship between the flavonoid content of grape cultivars and the fecundity of the avocado brown mite, *Oligonychus punicae* Hirst (Acari: Tetranychidae).

The life table parameters, particularly the intrinsic rate of natural increase (rm), are an essential parameter to assess and compare the population growth of a pest on different host plants [25,35]. In this study, the rm value was highest for *T. merganser* when it fed on *P. vulgaris* (0.391 day^−1^) compared to when it fed on *C. papaya* (0.371 day^−1^), *R. hybrida* (0.350 day^−1^), *C. annuum* var. *glabriusculum* (0.309 day^−1^), and *H. parvifolia* (0.271 day^−1^). Ullah et al. [8] reported that the rm of *T. merganser* was 0.379 (±0.005) day^−1^ and 0.279 (± 0.009) day^−1^ on *P. vulgaris* at 30 °C and 25 °C and 60−70% relative humidity and 16L:8D photoperiod, respectively. Reyes-Pérez et al. [12] documented that the rm was 0.21 day^−1^ on *C. papaya* at 27 °C and a relative humidity of 60 ± 2% and 14:10 h light: dark photoperiod. Treviño-Barbosa et al. [13] found that the intrinsic (infinitesimal) rate of increase (r) of *T. merganser* varied significantly by the host plant. The highest r of the red spider mite was observed in *C. papaya* (0.8482 ± 0.00), followed by *P. vulgaris* (0.7133 ± 0.01), *M. oleifera* (0.6146 ± 0.01), *C. annuum* var. *glabriusculum* (0.5568 ± 0.02), *H. parvifolia* (0.4068 ± 0.01), *P. tobira* (0.3379 ± 0.01), and *T. ahouai* (0.3421 ± 0.01). There was a broad variation in the rm values of *T. merganser* on the five host plants. Uddin et al. [17] mentioned that these variations may be due to the nutritional quality, chemical composition, and morphology of the leaf, as well as different experimental conditions. These characteristics of the host plant affect the development period, survival, and oviposition rate of tetranychids mites and consequently affect the intrinsic rate of natural increase, so rm adequately summarizes the physiological qualities of tetranychids mites in relation to their capacity to increase [35].

In conclusion, our results indicate that the demographic and biological parameters of *T. merganser* depend on the host plants. Among the tested host plants, *P. vulgaris* is most suitable for the population growth of *T. merganser* due to its shorter development period from egg to adult, the higher survival rate of immature mites, higher fecundity, and the fastest intrinsic rate of increase as compared to other host plants. This differential suitability of host plants for the red spider mite is an important factor to consider when developing biological control strategies into integrated pest management programs for *T. merganser*. Further research is necessary, including assessing the effect of host plants’ morphological, chemical, and nutritional characteristics on the biology and life table of *T. merganser*.

## Figures and Tables

**Table 1 insects-14-00473-t001:** Host plants used for the study of resistance to *Tetranychus merganser*.

Family	Scientific Name	Common Name	Location	Coordinates *	MASL *	Reported as Host Plant by
Caricaceae	*Carica papaya* L.	Papaya	Victoria City (semi-urban area)	23°46′22.3″ N 99°5′58.5″ W	256	Reyes-Pérez et al. [12]
Fabaceae	*Phaseolus vulgaris* L.	Bean	Victoria City (urban area)	23°45′28.84″ N 99°9′53.54″ W	297	
Rutaceae	*Helietta parvifolia* (Gray) Benth.	Barreta	Peregrina Canyon in Protected Natural Area “Altas Cumbres”, Victoria City	23°46′41″ N 99°12′12″ W	365	Monjarás-Barrera et al. [3]
Rosaceae	*Rosa* sp. L.	Rosebush	Victoria City (urban area)	23°46′19.2″ N 99°5′56.4″ W	286	Migeon and Dorkeld [5]
Solanaceae	*Capsicum annuum* L. var. *glabriusculum* (Dunal) Heiser y Pickersgill	Chili piquin	Protected Natural Area “Altas Cumbres”, Victoria City	23°41′52″ N 99°11′04″ W	411	Monjarás-Barrera et al. [4]

* Meters above sea level.

**Table 2 insects-14-00473-t002:** Developmental duration (days ± S.E.) from egg to adult of *Tetranychus merganser* females fed on five host plants at 28 ± 1 °C and 70–80% RH with a photoperiod of 12L:12D photoperiod.

Host Plant		Stages							
	n	Eggs	Larva	Protochrysalis	Protonymph	Deutochrysalis	Deutonymph	Teliochrysalis	Egg–Adult
*Phaseolus vulgaris*	72	3.97 ± 0.04 b*	1.13 ± 0.03 c	0.75 ± 0.03 b	0.74 ± 0.03 c	0.80 ± 0.03 c	0.92 ± 0.02 b	1.00 ± 0.02 b	9.32 ± 0.08 c
*Carica papaya*	62	3.97 ± 0.05 b	1.30 ± 0.04 b	0.75 ± 0.03 b	0.75 ± 0.03 c	0.83 ± 0.03 c	0.96 ± 0.02 b	1.02 ± 0.03 b	9.58 ± 0.10 c
*Capsicum annuum* var. *glabriusculum*	68	4.23 ± 0.04 a	1.77 ± 0.04 a	0.89 ± 0.03 a	0.91 ± 0.03 b	1.00 ± 0.10 ab	1.14 ± 0.03 a	1.20 ± 0.04 a	11.30 ± 0.16 a
*Rosa hybrida*	64	4.10 ± 0.05 a	1.40 ± 0.04 b	0.82 ± 0.03 ab	0.84 ± 0.03 bc	0.90 ± 0.03 bc	1.09 ± 0.03 a	1.13 ± 0.03 a	10.30 ± 0.09 b
*Helietta parvifolia*	63	4.18 ± 0.05 a	1.76 ± 0.03 a	0.91 ± 0.03 a	1.08 ± 0.03 a	1.08 ± 0.03 a	1.15 ± 0.04 a	1.21 ± 0.04 a	11.34 ± 0.10 a
Statistic (df = 4)		6.79	61.43	5.98	18.41	5.82	15.19	10.17	76.39
		*p* < 0.0001	*p* < 0.0001	*p* < 0.0001	*p* < 0.0001	*p* = 0.0002	*p* < 0.0001	*p* < 0.0001	*p* < 0.0001

* Within columns, different letters indicate significance differences among host plants (Tukey HSD test: *p* < 0.05).

**Table 3 insects-14-00473-t003:** Developmental duration (days ± S.E.) from egg to adult of *Tetranychus merganser* males fed on five host plants at 28 ± 1 °C and 70–80% RH with a photoperiod of 12L:12D photoperiod.

Host Plant		Stages							
	n	Eggs	Larva	Protochrysalis	Protonymph	Deutochrysalis	Deutonymph	Teliochrysalis	Egg–Adult
*Phaseolus vulgaris*	22	3.93 ± 0.04 b*	1.12 ± 0.05 c	0.75 ± 0.06 a	0.75 ± 0.06 c	0.81 ± 0.05 b	0.92 ± 0.04 b	1.02 ± 0.03 b	9.25 ± 0.11 c
*Carica papaya*	20	3.95 ± 0.04 b	1.30 ± 0.05 bc	0.74 ± 0.06 a	0.75 ± 0.06 c	0.84 ± 0.06 b	0.96 ± 0.03 b	1.00 ± 0.05 b	9.71 ± 0.11 c
*Capsicum annuum* var. *glabriusculum*	23	4.24 ± 0.08 a	1.78 ± 0.06 a	0.89 ± 0.05 a	0.92 ± 0.04 b	1.00 ± 0.06 ab	1.13 ± 0.06 a	1.21 ± 0.06 a	11.21 ± 0.17 a
*Rosa hybrida*	22	4.10 ± 0.04 ab	1.36 ± 0.07 b	0.82 ± 0.05 a	0.83 ± 0.06 bc	0.88 ± 0.05 ab	1.11 ± 0.05 a	1.12 ± 0.05 ab	10.22 ± 0.16 b
*Helietta parvifolia*	22	4.18 ± 0.07 a	1.75 ± 0.06 a	0.91 ± 0.05 a	1.09 ± 0.05 a	1.10 ± 0.05 a	1.13 ± 0.06 a	1.25 ± 0.07 a	11.50 ± 0.12 a
Statistic (df = 4)		5.96	23.70	2.18	6.38	4.11	4.49	3.79	45.59
		*p* = 0.0002	*p* < 0.0001	*p* = 0.0779	*p* < 0.0001	*p* = 0.0044	*p* = 0.0026	*p* = 0.0076	*p* < 0.0001

* Within columns, different letters indicate significance differences among host plants (Tukey HSD test: *p* < 0.05).

**Table 4 insects-14-00473-t004:** Hatchability of eggs, survival rates of immature stages, and sex ratio of *Tetranychus merganser* on different host plants.

Host Plant	Hatchability (%)	Survival Rate in Larvae (%)	Survival Rate in Protonymph (%)	Survival Rate in Deutonymph (%)	Survival Rate (Egg to Adult) (%)	Sex Ratio (% Female)
*Phaseolus vulgaris*	98.61	98.36	100.00	100.00	94.74	79.07
*Carica papaya*	98.39	96.72	98.28	96.43	83.87	78.79
*Capsicum annuum* var. *glabriusculum*	89.71	90.16	90.38	97.73	58.82	78.43
*Rosa hybrida*	96.88	95.16	98.25	96.36	79.69	76.12
*Helietta parvifolia*	88.89	91.07	89.58	87.80	53.97	77.27
χ^2^ (df = 4)	0.975	0.537	1.014	0.899	16.052	0.0769
*p*	0.914	0.970	0.908	0.925	0.003	0.9993

**Table 5 insects-14-00473-t005:** Hatchability of eggs and survival rates of immature stages of *Tetranychus merganser* male on different host plants.

Host Plant	Hatchability (%)	Survival Rate in Larvae (%)	Survival Rate in Protonymph (%)	Survival Rate in Deutonymph (%)	Survival Rate (Egg to Adult) (%)
*Phaseolus vulgaris*	100.00	95.45	100.00	100.00	81.82
*Carica papaya*	95.00	94.74	94.12	93.75	70.00
*Capsicum annuum* var. *glabriusculum*	91.30	95.24	94.74	93.75	60.87
*Rosa hybrida*	95.45	95.24	94.74	94.44	63.34
*Helietta parvifolia*	86.36	94.74	94.12	86.67	50.00
χ^2^ (df = 4)	1.111	0.0004	0.264	0.957	8.473
*p*	0.893	1.000	0.992	0.916	0.076

**Table 6 insects-14-00473-t006:** Reproduction parameters and adult longevity (mean ± SE) of *Tetranychus merganser* female on five host plants.

Host Plants	Pre-Oviposition	Oviposition	Post-Oviposition	Eggs/Female	Egg/♀/Day	Longevity
*Phaseolus vulgaris*	1.56 ± 0.07 ab*	14.6 ± 0.22 b	1.76 ± 0.05 a	125.40 ± 1.67 a	7.81 ± 0.06 a	18.04 ± 0.20 c
*Carica papaya*	1.68 ± 0.05 a	15.40 ± 0.21 b	1.80 ± 0.05 a	99.60 ± 1.24 b	7.25 ± 0.05 b	18.92 ± 0.32 b
*Capsicum annuum* var. *glabriusculum*	1.20 ± 0.06 cd	18.24 ± 0.22 a	1.38 ± 0.05 b	64.65 ± 0.87 c	5.28 ± 0.03 c	20.84 ± 0.15 a
*Rosa hybrida*	1.36 ± 0.06 bc	17.44 ± 0.20 a	1.50 ± 0.06 b	62.10 ± 1.21 c	4.93 ± 0.06 d	20.40 ± 0.22 a
*Helietta parvifolia*	1.1 ± 0.04 d	13.12 ± 0.19 c	0.66 ± 0.05 c	43.92 ± 0.073 d	4.00 ± 0.03 e	15.04 ± 0.17 d
Statistic (df = 4)	16.91	97.95	78.68	704.82	893.67	108.77
	*p* < 0.0001	*p* < 0.0001	*p* < 0.0001	*p* < 0.0001	*p* < 0.0001	*p* < 0.0001

* Within columns, different letters indicate significance differences among host plants (Tukey HSD test: *p* < 0.05).

**Table 7 insects-14-00473-t007:** Life table parameters of *Tetranychus merganser* on five host plants.

Host Plants	GRR Female/Female	R_O_ (Offspring)	GT (Days)	rc (Day^−1^)	rm (Day^−1^)	DT (Day)	λ (Day^−1^)
*Phaseolus vulgaris*	109.364 ± 0.928 a*	75.240 ± 0.935 a	11.060 ± 0.049 a	0.642 ± 0.005 a	0.391 ± 0.002 a	1.774 ± 0.010 a	1.478 ± 0.003 a
*Carica papaya*	101.430 ± 0.724 b	59.760 ± 0.699 b	11.022 ± 0.051 b	0.550 ± 0.006 b	0.371 ± 0.002 b	1.868 ± 0.011 b	1.449 ± 0.003 b
*Capsicum annuum* var. *glabriusculum*	77.457 ± 0.765 d	41.360 ± 0.534 c	12.037 ± 0.050 c	0.501 ± 0.003 c	0.309 ± 0.002 d	2.241 ± 0.015 d	1.362 ± 0.003 d
*Rosa hybrida*	84.041 ± 0.628 c	43.960 ± 0.563 c	10.817 ± 0.043 c	0.490 ± 0.005 c	0.350 ± 0.002 c	1.982 ± 0.013 c	1.419 ± 0.003 c
*Helietta parvifolia*	48.039 ± 0.420 e	22.840 ± 0.355 d	11.558 ± 0.082 d	0.439 ± 0.006 d	0.271 ± 0.003 e	2.561 ± 0.031 e	1.311 ± 0.004 e
Statistic (df = 4)	974.5	748.2	88.53	220.9	435.9	372.3	439.6
	*p* < 0.0001	*p* < 0.0001	*p* < 0.0001	*p* < 0.0001	*p* < 0.0001	*p* < 0.0001	*p* < 0.0001

* Within columns, different letters indicate significance differences among host plants (Tukey HSD test: *p* < 0.05).

## Data Availability

The data presented in this study are available on request from the corresponding author.

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
