# Peer review of "Life Table Parameters of Tetranychus merganser Boudreaux (Acari: Tetranychidae) on Five Host Plants"

_insects, 2023, doi:10.3390/insects14050473_

Round 1

Reviewer 1 Report

Reviewer

Manuscript ID: insects-2389210

Title: Life Table Parameters of Tetranychus merganser Boudreaux (Acari: Tetranychidae) on Five Host Plants

The authors present a study on the life history parameters of tetranychid mites. The biology and fertility life tables of Tetranychus merganser on five host plants (Carica papaya, Phaseolus vulgaris, Capsicum annuum var. glabriusculum, Helietta parvifolia, and Rosa sp.) were assessed under laboratory conditions at 28 ± 1 °C and 70–80% relative humidity (RH) with a photoperiod of 12:12 h (L:D). The authors discovered that the development period of immature females differed among the tested host plants and ranged from 9.32 days on P. vulgaris to 11.34 days on H. parvifolia. For immature males, it ranged from 9.25 days on P. vulgaris to 11.50 days on H. parvifolia. The female survival rate varied from 53.97% on H. parvifolia to 94.74% on P. vulgaris. The highest total fecundity rate was recorded on P. vulgaris (125.40 eggs/female) and lowest on H. parvifolia (43.92 eggs/female). The intrinsic rate of increase (rm) varied from 0.271 (H. parvifolia) to 0.391 (P. vulgaris). The net reproductive rate (RO) was higher on P. vulgaris than on the other host plants. The longest mean generation time (GT) was calculated on C. annuum var. glabriusculum and the shortest on Rosa hybrida. The demographic parameters suggest unsuitability of H. parvifolia as host for the development of red spider mite and best performance of T. merganser was on P. vulgaris.

The presented manuscript contains a study that is proper for Insects Journal. 

The information presented in the manuscript is original and documented.

  • The manuscript is sustained by a suitable literature.
  • The information from the manuscript is well structured.
  • The title of the manuscript reflects the content.
  • The aim of research is clearly defined.
  • The results and discussions are proper described and presented.
  • The results are comparable with other studied.
  • The data from results chapter are not repeated with those from figures and tables.
  • The tables and figures reveal properly the described results/data.

Despite all these positive aspects, some comments must be added:

1.      In the simple summary: line 17: the scientific names of the plants must be inserted.

2.      Abstract: line 28: It is Rosa sp. or Rosa hybrid as in lin 37?

3.      The key words: line 40: please arrange them in alphabetical order.

4.      Introduction: must be complete with additional information regarding the biology and ecology of the investigated species T. merganser. I saw that its biology is well described in the lines 60-82. But the authors, must to argued why they choose these abiotic conditions!  Why 30 degrees, why 70% relative humidity? Why only three months? Why 12L: 12 D? Please, see lines 97-98.

Lines 83-84: If they are a lot of studies on host plants and this mite species, why it is so important your study? Please, highlight which is novelty and originality of this study!

5.      On methods and material chapter, line 93, please mention the investigated stages of development! Line 94: please, insert the altitude!

Line 110: please, insert the references of presented methodology! The same for 2.3 chapter!

Line 120: What means (000)???

6.      Statistical analysis: Line 148: All these parameters must be explained and described at subchapter 2.4. Please use the same short names all over the manuscript (tables 3, 4, 5)!

7.      Chapter 3.3. Lines: 202-204. Check the line 148. On chapter 3.3 and in table 7 all parameters must to have the same short names and font format!  Here are six parameters, in table 7 and lines 148 are seven parameters!

8.      On conclusion: Please, insert a phrase with the practice role of your study in biological control!

9.      The manuscript is not written homogenous. The authors used common names or scientific names! The ife parameters are presented in many forms (RO, ro; GT or TG). Please, check these carefully!

  • References:

Please, check if the all references were found in the manuscript and vice versa.

Please, follow the instruction for the authors for references.

  • Check the English language again with a native English speaker!
  • All comments were inserting in the manuscript!

Author Response

Dear reviewer, we have read all your suggestions and the responses to them are presented below.

Suggestions

Reply

In line 17.

I consider that the scientific names must be insert first, and let them in brackets the common names!

Carica papaya L. (papaya), Capsicum annuum var. glabriusculum (Dunal) Heiser and Pickersgill (chili piquin), Helietta parvifolia (Gray) Benth. (barreta), Phaseolus vulgaris L. (bean), and Rosa hybrida L. (rosebush)

In line 28.

Rosa sp or Rosa hybrida

Rosa hybrida

In line 40.

Please, arrange them in alphabetic order!

demographic parameters; development; fecundity; red spider mite; survival

In line 83 and 84.

If they are a lot of studies on host plants and this mite species, why it is so important your study? Please, highlight which is novelty and originality of this study!

Besides, the mite fertility life-table parameters are used as an endpoint to reveal the susceptibility or resistance of host plants to the mite since host plants that support low mites population growth are important for integrated pest management [16].

In line 93.

Please, mention which stages!

we add

larvae, nymphs and adults

In line 95.

Altitude?

we add

411 meters above sea level

In line 97 and 98.

Please, give some scientific arguments for using these abiotic conditions!  Why 30 degrees, why 70% relative humidity ? Why only three months?

Why 12L:12 D?

I suggest to highlight these ecological preferences  with other studies in the introduction chapter!

We do not have scientific support regarding temperature (30 ± 2 °C) and relative humidity (70 ± 10%). These values were those registered in the greenhouse during the period in which the red spider was reared.

The photoperiod was established randomly, i.e., without any scientific reference. We have always used this data. Besides, we have not studied the effect of different photoperiods on the red spider mite for scientific reference.

Regarding the period of three months, it was the time in which the mite was raised, as could it be 1, 2, or 4 months. Also, it was the time the first author learned how to manage the mite colony.

In line 103.

What means MASL?

The meaning of MASL is: Meters above sea level

It was included as a footnote in the Table 1.

In line 110 and 111.

Please, insert the reference for methodology!

The reference is 14. (Treviño-Barbosa et al. 2022)

In line 112.

Please insert references for the used methodology! It must to be scientifically approved through other similar studies!

The references are found on line 118.

Which are:

Uddin et al. [16] and Gotoh and Gomi [17]

In line 120. (000) ???

It was removed (000)

In line 148.

All these parameters must be explained and described at subchapter 2.4.

Please used the same short names all over the manuscript!

In subchapter 2.4 the description of the parameters " gross reproductive rate (GRR) and growth capacity (rc)" are included.

These parameters were the only ones that were not described.

In Table 3.

Please, keep the same style as in table 2. If you use the scientific names, than do it all the manuscript!

In Tables 3, 4, 5, 6 and 7, the common names of the host plants were replaced by their scientific names.

In Table 6.

Correct: Longevity

In Table 6, we corrected the word Longevety to Longevity.

In line 202-204.

Check the line 148. On chapter 3.3  and in table 7 all parameters must to have the same short names and font format!

Here are six parameters, in table 7 and lines 148 are seven parameters!

In line 148, it has already been corrected and the seven parameters are described. Which coincides with subchapter 3.3 and Table 7.

In line 209.

RO or RO?

The correct is RO.

In line 211.

TG or GT ?

The correct is GT.

In line 311-312.

Please, insert a phrase with the practice role of your study in biological control!

This differential suitability of host plants for the red spider mite is an important factor to consider when developing biological control strategies into integrated pest management programs for T. merganser.

Reviewer 2 Report

Dear Authors,

The manuscript presents intersting findings regarding the biology of this emerging pest (Tetranychus merganser). It is well written and the results are clearly presented. I have only some minor corrections marked on the attached pdf file.

Sincerely

Author Response

Dear reviewer, we have read all your suggestions and the responses to them are presented below.

Suggestions

Reply

In line 51.

Replace with "7"

We replace the reference 8 with 7.

In line 64.

Replace with "to"

We replace the word “at” with “to”.

In line 65.

Change the order "different tested"

We changed "tested different temperatures" for "different tested temperatures"

In line 67.

Replace with "had"

We replace the word “has” with “had”.

In line 111.

Make "2" a superscript

We place the number 2 in superscript.

In line 118.

Replace with "mites"

We replace the word “mite” with “mites”.

In line 131.

Replace with "this"

We replace the word “one” with “this”.

In line 134.

Replace with "it"

We replace the word “he” with “it”.

In line 136.

Replace with "its"

We replace the word “her” with “its”.

In line 137.

Replace with "mites were transfered"

We replace the word “mites transfered” with “mites were transfered”

In line 139.

Remove "the female of"

We removed the word " the female of "

In line 144.

Replace with "18"

We corrected the reference number. Reference 19 was changed to 18.

In line 146.

Replace with "18"

We corrected the reference number. Reference 19 was changed to 18.

In line 150.

Remove “period”

We removed the word "period"

In line 156.

Replace with "tested"

We corrected the word “tasted” with “tested”.

In line 156.

Replace with "egg"

We corrected the word “eggs” with “egg”.

In line 156.

Replace with "lava"

We corrected the word “larval” with “larva”.

In line 157.

Replace with “protonymph”

We corrected the word “protonymphals” with “protonymph”.

In line 157.

Replace with “deutonymph”

We corrected the word “deutonymphs” with “deutonymph”.

In line 158.

Replace with “tested”

We corrected the word “tasted” with “tested”.

In line 159.

Replace with “time”

We corrected the word “times” with “time”.

In line 166 to 168.

Rephrase sentence "Regarding T. merganser male, the mean developmental time of egg, larva, protonymph, deutochrysalis, deutonymph, and theliochrysalis differed significantly between the tested host plant"

We accept the reformulation of the sentence.

“Regarding T. merganser male, the mean developmental time of egg, larva, protonymph, deutochrysalis, deutonymph, and theliochrysalis differed significantly between the tested host plant”

In line 169.

Replace with “time”

We corrected the word “times” with “time”.

In line 178.

Replace with "did"

We corrected the word “does” with “did”.

In line 203.

Make "-1" a superscript

We did "-1" a superscript

In line 208.

Use italics for T. merganser

We used italics for T. merganser

In line 208.

Use italics for P. vulgaris

We used italics for P. vulgaris

In line 225.

Replace with "developmental"

We corrected the word “development” with “developmental”.

In line 227.

Replace with "developmental"

We corrected the word “development” with “developmental”.

In line 228.

Replace with "duration of development"

We corrected the word “development times” with “duration of development”.

In line 270.

Replace "vulgaris"

We corrected the word “phaseolus” with “vulgaris”.

In line 270.

Replace with "secondary"

We corrected the word “secundary” with “secondary”.

In line 284.

Replace "vulgaris"

We corrected the word “phaseolus” with “vulgaris”.

In line 294.

Replace "vulgaris"

We corrected the word “phaseolus” with “vulgaris”.

In line 297.

Replace "vulgaris"

We corrected the word “phaseolus” with “vulgaris”.

In line 397.

Remove the number 28.

We removed the number 28 from the reference 28. This number was repeated.

Round 2

Reviewer 1 Report

I still insist!  If they are a lot of studies on host plants and this mite speci, why it is so important your study? Please, highlight which is novelty and originality of this study!

  • I strongly advice authors to highlighted the novelty of the present research.
  • I discover another paper with similar research:  “The Resistance of Seven Host Plants to Tetranychus merganser Boudreaux (Acari: Tetranychidae)”. Authors: Guadalupe Treviño-Barbosa, Salvador Ordaz-Silva, Griselda Gaona-García, Agustín Hernández-Juárez, Sandra Grisell Mora-Ravelo and Julio César Chacón Hernández, published in 2022. This paper present a similar research subject, with the same species Tetranychus merganser on seven host plants (4 of them are common with your study). Five life parameters investigated in 2022 (DT, the finite rate of increase, grow rate, finite grow rate, fecundity) are similar with your study! Considering all these informations. I strongly recommend highlighting the novelty of the present research. Please, mention that the life table parameters are most detailed analyzed, taking account all immature stages, more detailed reproduction parameters, survival rates of all immature stages, differences between male and females. If the authors don’t highlight these novelties, the quality of the present will drastically decrease.
  • References: the authors inserted 34 references. From all these 8 references are self-citation (no 3, 4, 6, 7, 11, 13, 14, 15). These mean almost 25% from all references. Is too much, please reconsider this!
  • At discussion chapter I recommend a more detailed comparative analysis with the study made in 2022.
  • Be careful with the English spelling of words. Ex: line 30 Rosahybrida in one word or line 124 – Tmerganser.

Minor editing of English language required

Author Response

Dear reviewer, we have read your suggestions and observations made on the manuscript. We made all your suggestions.

Reviewer Tip

Reply

I strongly recommend highlighting the novelty of the present research. Please, mention that the life table parameters are most detailed analyzed, taking account all immature stages, more detailed reproduction parameters, survival rates of all immature stages, differences between male and females. If the authors don’t highlight these novelties, the quality of the present will drastically decrease.

The following lines were written.

Although Treviño-Barbosa et al. [13] have already reported some demographic and biological parameters of T. merganser, such as the intrinsic (infinitesimal) rate of increase (r), the finite growth rate (λ), doubling time (DT), and oviposition on C. papaya, P. vulgaris, C. annuum var. glabriusculum, and H. parvifolia. These results approximate the population growth of the red spider mite when it feeds on a host plant since r is a crude rate and establishes a foundation for examining population growth patterns over short periods [14,15], such as the study by Treviño-Barbosa et al. [13]. They calculated r on the fourth day. To have a clear and systematic picture of the specific age at birth (mx) and survival (lx) of the red spider mite population [15,16], we analyze in more detail the developmental time from egg to adult for both female and male, the pre-oviposition, oviposition, and post-oviposition periods, fecundity rate, survival rates of all immature stages and differences between male and female, and several life table parameters of T. merganser.

References: the authors inserted 34 references. From all these 8 references are self-citation (no 3, 4, 6, 7, 11, 13, 14, 15). These mean almost 25% from all references. Is too much, please reconsider this!

We remove references 7, 13 and 15.

7. Villagran-Mancilla, C.; Chacón-Hernández, J.C.; Delgadillo‑Ángeles, J.L; Hernández-Juárez, A.; Mora-Ravelo, S.G.; Ordaz-Silva, S. Phytophagy and predatory behavior of Caliothrips phaseoli (Thysanoptera: Thripidae) on bean foliage discs with Tetranychus merganser (Acari: Tetranychidae) eggs. Arthropod-Plant Interactions. 2023, 17: 217–224. https://doi.org/10.1007/s11829-023-09949-w

13. Chacón-Hernández, J.C.; Ordaz-Silva SMireles-Rodriguez, E.; Rocandio-Rodríguez, M.; López-Sánchez, I.V.; Heinz-Castro, R.T.Q.; Reyes-Zepeda, F.; Castro-Nava, S. Resistance of wild chili (Capsicum annuum L. var. glabriusculum) to Tetranychus merganser1 Boudreaux. Southwest. Entomol. 2020, 45, 89–98. https://doi.org/10.3958/059.045.0110

15. Rocandio-Rodriguez, M.; Torres-Castillo, J.A.; Juárez-Aragón, M.C.; Chacon-Hernandez, J.C.; Moreno-Ramirez, Y.d.R.; Mo-ra-Ravelo, S.G.; Delgado Martinez, R.; Hernandez-Juarez, A.; Heinz-Castro, R.T.Q.; Reyes-Zepeda, F. Evaluation of resistance of eleven maize races (Zea mays L.) to the red spider mite (Tetranychus merganser, Boudreaux). Plants 2022, 11, 1414. https://doi.org/10.3390/plants11111414

References were incorporated

7. López-Bautista, E. Incidence of Damage and Control Strategies of Tetranychus merganser in the Papaya Crop (Carica papaya L.). Ph.D. Thesis, Postgraduate College Campus Montecillo, Texcoco, Mexico, 2014.

14. Carey, J.R. Applied Demography for Biologists with Special Emphasis on Insects, Oxford. New York, USA, 1993; p. 206.

15. Smith, T.M., Smith, R.L. Elements of Ecology. 9th. edition; pearson. Essex, England, 2015; p. 683.

16. Carey, J.R. Insect biodemography. Annu. Rev. Entomol. 2001. 46:79–110. https://doi.org/10.1146/annurev.ento.46.1.79

At discussion chapter I recommend a more detailed comparative analysis with the study made in 2022.

The following paragraphs were incorporated into the discussion chapter.

Treviño-Barbosa et al. [13] documented that the mean number of T. merganser eggs laid per female on C. papaya (9.39±0.41) was higher than on P. vulgaris (5.54±0.37), M. oleifera (3.97±0.16, C. annuum var. glabriusculum (2.58±0.14), H. parvifolia (1.57±0.05), P. tobira (1.00±0.04), and T. ahouai (1.06±0.05).

Treviño-Barbosa et al. [13] found that the intrinsic (infinitesimal) rate of increase (r) of T. merganser varied significantly by the host plant. The highest r of the red spider mite was observed in C. papaya (0.8482±0.00), followed by P. vulgaris (0.7133±0.01), M. oleifera (0.6146±0.01), C. annuum var. glabriusculum (0.5568±0.02), H. parvifolia (0.4068±0.01), P. tobira (0.3379±0.01), and T. ahouai (0.3421±0.01).

Be careful with the English spelling of words. Ex: line 30 Rosahybrida in one word or line 124 – Tmerganser.

We corrected the words "Rosahybrida" for "Rosa hybrida"

And

“Tmerganser” for “T. merganser